# An Opposition-Based Learning CRO Algorithm for Solving the Shortest Common Supersequence Problem

**DOI:** 10.3390/e24050641

**Published:** 2022-05-03

**Authors:** Fei Luo, Cheng Chen, Joel Fuentes, Yong Li, Weichao Ding

**Affiliations:** 1School of Information Science and Engineering, East China University of Science and Technology, Shanghai 200237, China; luof@ecust.edu.cn (F.L.); cheng.chen2@yintech.cn (C.C.); 2Department of Computer Science and Information Technologies, Universidad del Bío-Bío, Chillán 3780000, Chile; jfuentes@ubiobio.cl

**Keywords:** chemical reaction optimization, opposition-based learning, shortest common supersequence, heuristic algorithm, NP-hard

## Abstract

As a non-deterministic polynomial hard (NP-hard) problem, the shortest common supersequence (SCS) problem is normally solved by heuristic or metaheuristic algorithms. One type of metaheuristic algorithms that has relatively good performance for solving SCS problems is the chemical reaction optimization (CRO) algorithm. Several CRO-based proposals exist; however, they face such problems as unstable molecular population quality, uneven distribution, and local optimum (premature) solutions. To overcome these problems, we propose a new approach for the search mechanism of CRO-based algorithms. It combines the opposition-based learning (OBL) mechanism with the previously studied improved chemical reaction optimization (IMCRO) algorithm. This upgraded version is dubbed OBLIMCRO. In its initialization phase, the opposite population is constructed from a random population based on OBL; then, the initial population is generated by selecting molecules with the lowest potential energy from the random and opposite populations. In the iterative phase, reaction operators create new molecules, where the final population update is performed. Experiments show that the average running time of OBLIMCRO is more than 50% less than the average running time of CRO_SCS and its baseline algorithm, IMCRO, for the desoxyribonucleic acid (DNA) and protein datasets.

## 1. Introduction

As a well-known non-deterministic polynomial hard (NP-hard) problem [1], the shortest common supersequence (SCS) problem has been widely applied in real life and in many bioinformatics and computer science fields, including pattern finding [2], multiple sequence alignment problems [3], deoxyribonucleic acid (DNA) sequencing [4], data compression [5], and artificial intelligence (AI) planning [6]. To find the optimal solution for the SCS problem many different studies have been proposed, including artificial bee colonies (ABC) [7], ant colony optimization (ACO) [8], enhanced beam search (IBS) [9], deposition and reduction (DR) [10], and the CRO_SCS (chemical reaction optimization (CRO) for the SCS problem) algorithm [11].

Compared to other heuristic algorithms, CRO algorithms have achieved better results in solving the SCS problem. The CRO algorithm is inspired by the process of obtaining low-potential energy molecules from chemical reactions in nature. The algorithm calls the solution of the problem a molecule, which is obtained from an iterative process in which four reaction operators are used to generate new molecules performing local and global search. To solve the SCS problem, Saifullah et al. proposed the CRO_SCS algorithm, a CRO-based proposal with the inclusion of a check and repair function to repair molecules after iterations [11]. The operator checks each character of the generated superstring and records the number of illegal characters. If the number of violations exceeds a certain standard, the string is discarded. If the number of violations is less than a predefined value, it is repaired. Compared to AOC, ABC, IBS and DR, CRO_SCS is able to shorten the average length of SCS and reduce the average running time.

In order to further improve the performance of the CRO algorithm in solving the SCS problem, a new algorithm based on CRO_SCS, the IMCRO algorithm, was proposed in [12]. It introduced the circular shift operator and the two-step crossover operator in the decomposition reaction and the intermolecular collision reaction. Compared to CRO_SCS, the average length of the shortest common superstring calculated by IMCRO is shorter by 1.02 characters on average.

Even though these CRO-based algorithms have shown important performance gains, they can only perform iterative searches based on the current solution. On the one hand, they are affected by the randomness of the initial population, while on the other they face the problems of slow convergence and premature convergence [13]. The opposition-based learning (OBL) mechanism has arisen as a powerful technique for overcoming these problems. In OBL, the current solution and the reverse solution are examined at the same time during the search process, which improves the diversity of the population and at the same time expands the search range of the algorithm. By doing this the algorithm has more opportunities to escape from the current search area, improving its global search ability. This can effectively solve the premature optimization problem of CRO-based algorithms and achieve convergence in fewer iterative steps, which accelerates the convergence speed [14].

This article introduces the OBL into IMCRO, proposing an opposition-based learning CRO algorithm dubbed OBLIMCRO. The main contribution of this paper is as follows:Proposing an OBL function to solve the SCS problem by generating an opposite solution corresponding to the original one. When a new solution is produced, its reverse solution will be generated by the OBL function.Introducing the OBL mechanism into the construction of the population by mixing random molecules and opposite molecules, which can be used to optimize the quality of the population and enrich the diversity of the population.Proposing an improved CRO-based algorithm with the OBL mechanism in the initialization stage and the iteration stage, resulting in speeding up the convergence and solving the premature trap.

The rest of the paper is organized as follows: related works are summarized in Section 2; the design details of OBLIMCRO are presented in Section 3, where the OBL mechanism for population generation and updating is depicted; to illustrate the performance of OBLIMCRO, an evaluation strategy is defined and the resulting experiments and analysis are shown in Section 3.2.3; finally, the conclusions of the paper are drawn in Section 5.

## 2. Related Work

David Maier first defined the SCS problem in 1976 [15], which was proved to be NP-complete for sequences over an alphabet of size five. It has been widely used in bioinformatics and computer science. Different algorithms have been proposed to solve the SCS problem, including heuristic algorithms such as [8], ABC [7], IBS [9], DR [10], and CRO_SCS [11].

The CRO algorithm was originally proposed by Lam and Li in 2010 [16], and was inspired by the process of chemical reactions. In short, it is based on the fact that a chemical reaction undergoes sub-reactions where certain intermediate states correspond to a reaction. The molecule becomes more stable when the molecule’s energy is lower than it was in the prior state.

Following the CRO principle, the CRO_SCS algorithm effectively solves the SCS problem. Compared to other heuristic algorithms, the average length of the shortest common superstring obtained by CRO_SCS is smaller under the same conditions. However, as a heuristic algorithm it faces the problems of slower convergence speed and premature maturity. The IMCRO algorithm [12] is an improved algorithm based on CRO_SCS. It introduces the two-step crossover and circular shift operators for the intermolecular collision and the decomposition reactions, respectively. The use of the reaction operator expands the degree of molecular structure change, improving overall algorithm performance. The premature problem is addressed by the algorithm’s local search and global search capabilities. Compared to CRO_SCS, the average length of the shortest common superstring output by IMCRO is reduced by 1.02 characters.

By combining the CRO with other heuristic algorithms, many novel CRO-based algorithms have been proposed, including random molecule-based chemical reaction optimization (RMCRO) [17], particle swarm and CRO (HP-CRO) [18], employed bee operator-based CRO (EBCRO) [19] and bat mutation-based CRO (BMCRO) [20].

OBL is an optimization strategy used in machine learning that was proposed by Tizhoosh in 2005. The main idea of the algorithm is to simultaneously evaluate both the current solution and its opposite solution during the search process. The algorithm then selects the solution closer to the optimal solution from among both the current and opposite solutions. It then performs subsequent searches, reaching the optimal solution in fewer iterations and speeding up the search process.

Meta-heuristic algorithms use OBL to obtain better results in terms of optimizing the population quality and accelerating the speed of convergence. The work presented in [21] combines the harmony search algorithm (HS) and the OBL mechanism to construct the opposition-based learning global harmony search algorithm (OLGHS). The OBL mechanism participates in the establishment of the initial harmony library, and is considered in the initialization process. For forward and reverse solutions, it chooses the proper solution to establish the initial harmony library. This is in order to ensure that the quality of the initial harmony library is appropriate as well as to solve the premature problem by optimizing the quality of the harmony library.

Several novel CRO algorithms combined with OBL have been proposed. Examples include the quasi-opposition learning chemical reaction optimization algorithm (QOCRO) [13], used to solve the reaction energy scheduling problem, and the elite opposition-based learning chemical reaction difference (EOLCRDE) optimization algorithm [22], used to solve the power dispatch problem.

To the best of our knowledge, the OBL mechanism has not previously been used to solve SCS problems. This work introduces OBL in IMCRO (our previous work) to solve the aforementioned problem.

## 3. OBLIMCRO

### 3.1. Opposition-Based Learning

The OBL strategy included in IMCRO is intended to generate “high-quality” solutions to accelerate the iteration speed. In each iteration of the heuristic algorithm, when a new solution (e.g., the forward solution) is produced its reverse solution is provided by OBL as well. The forward solution and the reverse solution are compared and the better one joins the population. This selection process is performed before starting the next iteration, and is equivalent to expanding the search range.

Notice that by applying the OBL mechanism the quality of the population is optimized and the diversity of the population is enriched. This results in improving the algorithm’s global search ability, allowing it to avoid falling into the local optimum trap.

### 3.2. Oblimcro Framework

The same as IMCRO, OBLIMCRO has three stages, initialization, iteration, and finalization, as shown in Figure 1. The general procedure is described in Algorithm 1, while the stages of the framework are detailed in the following subsections.
**Algorithm 1** Framework of OBLIMCRO**Input:** Sets of populations.
  1:Initialization of parameters, such as buffer, Initial KE, PopSize, KELossRate, MoleColl, β and α.  2:Create PopSize number of random molecules  3:Calculate opposite molecules of random molecules  4:Mix random molecules and opposite molecules  5:Select PopSize molecules with lowest potential energy from the mix set  6:**while** the stopping criteria is not met **do**  7:  Generate t∈[0,1]  8:  **if** (NumHit−MinHit)>α **then**  9:    Randomly select one molecule m10:    **if** (NumHit−MinHit)>α **then**11:      Trigger Decomposition12:    **else**13:      Trigger On-wall Ineffective Collision14:    **end if**15:  **else**16:    Randomly select two molecules $m_{1}$ and $m_{2}$17:    **if** KE≤β **then**18:      Trigger Synthesis19:    **else**20:      Trigger Inter-molecular ineffective collision21:    **end if**22:  **end if**23:  Check for new better solution24:  Calculate opposite molecule25:  Choose the solution with lower PE from the solution and its opposite molecule26:  Update population with the chosen solution27:**end while**
**Output:** Best solution.


#### 3.2.1. Initialization Stage

Initialization is the first stage of OBLIMCRO, where the molecules are initialized. Initial values of molecules include PopSize, KELossRate, MoleColl, buffer, Initial KE, α, and β. Therein, Popsize is the number of all feasible solutions, KELossRate is the maximum percentage of KE reduction, MoleColl is used to determine whether the chemical reaction is inter-molecule or uni-molecule, KE is the initial kinetic energy of the molecule, and α and β are the thresholds for intensification and diversification.

Molecules in the initial population group are generated with OBL. First, the initial molecule group, $\mathrm{RX}=\left\{x_{1},x_{2},\ldots ,x_{n}\right\}$, is randomly generated; then, its opposite group, $\mathrm{OX}=\left\{ox_{1},ox_{2},,ox_{n}|ox_{i}=\mathrm{obl}\left(x_{i}\right)\right\}$, is generated with the function obl based on the OBL mechanism. Afterwards, the random and opposite group are mixed as MX, where MX=RX∪OX. The individuals in MX are sorted according to their potential energy and the PopSize molecules with the lowest potential energy are selected to form the initial population, denoted as IX. Figure 2 shows an example of the initialization process, where the number of molecules in RX, OX, MX, and IX is PopSize.

The construction of RX, OX, MX, and IX takes place in the initialization stage. RX is constructed with a random inserting operation, as depicted in Figure 3. Therein, the initial superstring C is randomly generated as an array the elements of which are composed of letters in the string alphabet. The element of $S_{i}$ is randomly inserted into C, where $S_{i}$ is one string of the input string set S and 1≤i≤PopSize. This random inserting operation is performed on all strings in S to obtain the initial population with PopSize molecules. For further processing, each supersequence is encoded as a set of integer values. For instance, Σ is an alphabet set and Σ={a,c,g,t}. It is encoded as I={0,1,2,3}, where each element of Σ is encoded as an integer in I in the corresponding position of the set. Then, the set {0,1,2,3,2,1,3,0} can represent the supersequence E={a,c,t,g,t,c,g,a}, as shown in Figure 4.

OX is obtained by the OBL mechanism after the initialization of RX. For each molecule X with n elements in RX, where $X=\left\{x_{1},x_{2},\ldots ,x_{n}\right\}$, its opposite molecule OX is obtained using the function obl(), as defined in Formula (Equation 1). Therein, the function o(x) obtains the inverse number of the integer x. For a real number x the feasible region of which is [a,b], its inverse number ox is defined in Formula (Equation 2). Taking a molecule X={0,1,2,3,2,1,3,0} with a feasible region [0,3] as an example, according to Formula (Equation 1), the opposite molecule is OX={3,2,1,0,1,2,0,3}.
(1)$$\mathrm{OX}=\mathrm{obl}\left(X\right)=\mathrm{obl}\left(\left\{x_{1},x_{2},...,x_{n}\right\}\right)=\left\{o\left(x_{1}\right),o\left(x_{2}\right),...,o\left(x_{n}\right)\right\}$$
(2)ox=o(x)=a+b−x

After obtaining RX and OX, MX is obtained by performing the collective union operation on RX and OX, where $\mathrm{MX}=\mathrm{RX}\cup \mathrm{OX}=\left\{x_{1},x_{2},\ldots ,x_{n},ox_{1},ox_{2},\ldots ,ox_{n}\right\}$ with the scale as 2*PopSize. The molecules are sorted according to the molecular potential energy of the molecules in MX. Half of the molecules in MX with the lowest potential energy are selected to construct IX, which is the initial molecule population.

#### 3.2.2. Iteration Stage

The iteration stage consists of two subtasks, reaction and repair. There are four main operators used in the action subtask: on-wall ineffective collision, decomposition, inter-molecular ineffective collision, and synthesis. On-wall ineffective collision is realized by the collision of a single molecule. Then, one element of the molecule is randomly selected and changed to form a new molecule. The operator is used for local search. Decomposition refers to a molecule colliding with the wall and splitting into two or more molecules, and is usually used for global search. Inter-molecular ineffective collision is similar to on-wall ineffective collision. Synthesis is the combination of two molecules into a new molecule, which is equivalent to the inverse reaction of decomposition.

The four operators fall into two types, uni-molecule reaction and inter-molecule reaction. Therein, uni-molecule reactions include on-wall ineffective collision and decomposition, while inter-molecule reactions include inter-molecular ineffective collision and synthesis. The four operators in this stage are the same as those defined in [12].

At the start of each iteration step, a parameter t is randomly generated. This parameter determines whether uni-molecule reactions or inter-molecule reactions are triggered. In particular, uni-molecule reactions are triggered when t>MoleColl; otherwise, inter-molecule reactions are triggered. The specific operator is further determined according to values of α and β. In uni-molecule reactions, decomposition is triggered if (NumHit−MinHit)>α; otherwise, on-wall ineffective collision is triggered. In inter-molecule reactions, synthesis is triggered if KE≤β; otherwise, inter-molecular ineffective collision is triggered.

Through the processing of the chemical operators a new molecule, $X_{i}^{\prime }$, is created based on the molecule $X_{i}$ and an updating process is triggered, as shown in Algorithm 2. First, the opposite molecule, $OX_{i}^{\prime }$, is created according to Formula (Equation 1). The potential energy of $X_{i}^{\prime }$ and $OX_{i}^{\prime }$ are compared and the molecule with lower potential energy is chosen to be returned to the initial population for updating.
**Algorithm 2** Update based on OBL.**Input:** new molecule $X_{i}^{\prime }$. $OX_{i}^{\prime }$ = obl($X_{i}^{\prime }$);
  2:PE($X_{i}^{\prime }$) = f($X_{i}^{\prime }$);  PE($OX_{i}^{\prime }$) = f($OX_{i}^{\prime }$);  4:**if** PE($X_{i}^{\prime })<PE(OX_{i}^{\prime }$) **then**;    $UX_{i}^{\prime }=X_{i}^{\prime }$;  6:**else**    $UX_{i}^{\prime }$ = $OX_{i}^{\prime }$;  8:**end if**  **if**
$UX_{i}^{\prime }$ matches stopping criteria **then**10:  output $UX_{i}^{\prime }$;  **else**12:  put $UX_{i}^{\prime }$ into IX to update the initial group;  **end if**


Similar to that in IMCRO, a repair function to check this solution is triggered when a solution is obtained. If the requirement of the problem is satisfied, the obtained solution is repaired by a repair algorithm until the termination condition is met. Then, it enters the final stage, where the best solution is output and returned when the stopping criteria are met; otherwise, iteration is triggered again and the reactions are repeated.

#### 3.2.3. Finalization

At the end of each iteration, termination conditions are checked and a new molecule is obtained through the OBL-based updating process (Algorithm 2). If the conditions are met, either a new molecule is output as the result or the iteration continues. There are two termination conditions. One is that the potential energy of the new molecule is less than the specified threshold, and the other is that the number of iterations reaches a certain upper limit. If either of the two conditions is met, it is deemed to have met the termination condition.

## 4. Experiments and Evaluation

In order to evaluate the performance and efficiency of OBLIMCRO for solving the SCS problem, we carried out a set of experiments where OBLIMCRO was compared to IMCRO and CRO_SCS, as well as to heuristic algorithms such as ACO, IBS, and DR.

### 4.1. Configuration of Experiments

The execution environment for the compared algorithms was a Windows-based personal computer with an i5-4210U CPU and 4.0GB RAM. We adopted the same parameters for the experiments as those in IMCRO [12]. In particular, MoleColl was 0.2, KELossRate was 0.6, and PopSize was 20.

The datasets were the same as those used in IMCRO [12], including a random DRM dataset with fifteen instances and a real DRL dataset with eleven instances. There were fifteen DNA sequences in the random DRM dataset, with each instance containing four different types of characters (Σ=4). There were six DNA sequences and five protein sequences in the real DRL dataset. There were twenty different types of characters (Σ=20) in each protein sequence.

In the experiment, each compared algorithm was run twenty times with the different datasets. For each instance we obtained 200 different results, including the length of SCS and the execution time. The average length and the average execution time for the SCS were obtained from these 200 results. After repeating the process described above, results such as the average SCS length, represented as *L*, and the average execution time, represented as *T*, were obtained. The value of *L* of the specialized algorithm *alg* for the instance *ins* is denoted as $L_{alg}\left(ins\right)$. Meanwhile, the value of *T* of the specialized algorithm *alg* for the instance *ins* is denoted as $T_{alg}\left(ins\right)$; alg is one of the compared algorithms, e.g., *alg* ∈$\left\{DR,IBS,ACO,CRO_{-}SCS,IMCRO,OBLIMCRO\right\}$, and ins is one type of the datasets, e.g., *ins*
∈DRM∪DRL.

The maximum number of iterations for the CRO operations and the current potential energy exceeding the threshold were the two stopping criteria for the execution of the compared algorithms. Specifically, the maximum number of iterations was 500 and the threshold of the potential energy was specialized according to the structure of the particular instance. As depicted in Algorithm 1, if one of the two criteria were satisfied, the algorithm was stopped and the final solution was output.

### 4.2. Results and Analysis

The objective of the compared algorithms was to obtain the shortest average SCS length and the minimum average execution time. Table 1, Table 2, Table 3 and Table 4 show the results of the algorithms with these two types of datasets. Therein, *n* is the instance’s string number and *k* is the string length. The emboldened values in each table are the best results.

Table 1 shows that $L_{OBLIMCRO}\left(ins\right)<L_{alg}\left(ins\right)$, where ins∈DRM and $\;\mathrm{alg}\;\in \left\{DR,IBS,ACO,CRO_{-}SCS,IMCRO\right\}$. Table 2 shows that $T_{OBLIMCRO}\left(ins\right)\leq T_{alg}\left(ins\right)$, where ins∈DRM and *alg*
$\in \left\{DR,IBS,ACO,CRO_{-}SCS,IMCRO\right\}$. The results show that when compared with DR, IBS, ACO, CRO_SCS and IMCRO, OBLIMCRO obtained the minimum average SCS length and minimum average execution time.

Table 3 shows that $L_{OBLIMCRO}\left(ins\right)<L_{alg}\left(ins\right)$, where ins∈DRL except ins∈{PROT−3,PROT−4,PROT−5} and $\;\mathrm{alg}\;\in \left\{DR,IBS,ACO,CRO_{-}SCS,IMCRO\right\}$. Meanwhile, $L_{OBLIMCRO}\left(PROT-3\right)$, $L_{OBLIMCRO}\left(PROT-4\right)$, and $L_{OBLIMCRO}\left(PROT-5\right)$ are closest to $L_{IMCRO}\left(PROT-3\right)$, $L_{IMCRO}\left(PROT-4\right)$, and $L_{CRO\_SCS}\left(PROT-5\right)$, respectively, which are the smallest values. Table 4 shows that $T_{OBLIMCRO}\left(ins\right)\leq T_{alg}\left(ins\right)$, where ins∈DRL and alg∈{DR,IBS,ACO,CRO_SCS,IMCRO}.

The results shown in Table 1 and Table 3 indicate that OBLIMCRO obtains the minimum value in both average SCS length and average execution time for DNA instances in real datasets. However, OBLIMCRO does not considerably overwhelm IMCRO and CRO_SCS for protein instances, while OBLIMCRO, IMCRO and CRO_SCS achieve smaller values of the average SCS length in comparison with the other algorithms. In particular, in comparison with IMCRO OBLIMCRO can reduce L for DNA instances, as shown in Formula (Equation 3), where RC is the average reduced value. In DRM, RC with OBLCRO is 2.11, where num = 15. On the other hand, RC with OBLCRO is 2.24 in DRL, where num = 6.

Except for the reduction in average SCS length, the reduction in average running time is better. In the random set, the average running time of the OBLIMCRO algorithm is the lowest among the three CRO algorithms. Compared with the CRO_SCS algorithm, after using the OBLIMCRO algorithm the average running time is shortened by 52.8%. Compared with IMCRO, after using the OBLIMCRO algorithm the average running time is shortened by 50.9%. In the real set, a substantial improvement in the average running time was found. Compared with CRO_SCS, the average running time of OBLIMCRO is shortened by 54.3%, and compared with IMCRO, the running time of OBLIMCRO is shortened by 50.3%. It can be seen that whether using experimental data in a random set or using a real set, after using OBLIMCRO the running time can be shortened by more than half when compared with CRO_SCS and IMCRO. The average reduction in average running time, ΔT, for *T* can be obtained using Formula (Equation 4).
(3)$$RC=\frac{\sum_{i=1}^{num}\left(L_{IMCRO/CRO\_SCS}\left(DNA-i\right)-L_{OBLIMCRO}\left(DNA-i\right)\right)}{num}$$
(4)$$T=\frac{T_{IMCRO/CRO\_SCS}\left(i\right)-T_{OBLIMCRO}\left(i\right)}{T_{LMCRO/CRO\_SCS}\left(i\right)}$$

To prove the credibility of the results above, we applied a statistical significance test to $L_{OBLIMCRO}$ on $L_{CRO\_SCS}$ and $L_{IMCRO}$ using *t*-tests [23]. The *p*-value of the *t*-test was $2.83\times 10^{-5}$ and 0.0013 for the random datasets (Table 1), respectively, and 0.04 and 0.0164 for the real datasets (Table 3), respectively. All of these *p*-values are smaller than 0.05, which means that the difference in the average SCS length between OBLIMCRO and IMCRO or CRO_SCS is statistically significant. The results of the *t*-tests statistically verify the credibility of the results, showing that OBLIMCRO can efficiently reduce the average SCS length for most of the datasets in comparison with IMCRO and CRO_SCS.

Next, we applied a statistical significance test to $T_{OBLIMCRO}$ on $T_{CRO\_SCS}$ and $T_{IMCRO}$ using *t*-tests, as well. For the random dataset (Table 1), the *p*-value of the *t*-test was 0.12 and 0.13 respectively, while for the real datasets (Table 3) it was 0.021 and 0.024 respectively. As all of these *p*-values are smaller than 0.05, the difference in the average running time between OBLIMCRO and IMCRO or CRO_SCS is statistically significant. The results of the *t*-tests statistically verifies the credibility of te results, showing that in comparison with IMCRO and CRO_SCS, OBLIMCRO can efficiently reduce the running time required to resolve the SCS problem.

The number of iterations required to reach the best solution with the CRO algorithms is shown in Table 5 and Table 6, and is related to the PE threshold of each instance. For the three CRO algorithms, the number of CRO operations per iteration is less than 500, which is configured as the maximum iterations. This indicates that the three CRO algorithms converge invariantly.

## 5. Conclusions

We proposed a novel algorithm by introducing an OBL mechanism into IMCRO to solve the SCS problem. The proposed algorithm, named OBLIMCRO, aims to tackle common problems found in state-of-the-art algorithms for SCS. The OBL mechanism participates in the process of constructing and updating the population, which optimizes the quality of the population that is used to obtain the final solution. Experimental results show that for DNA proteins in both random and real datasets, the average running time is reduced and the average length required to solve the SCS is shortened. The results indicate that with the help of OBL, OBLIMCRO can achieve more optimal solutions with a faster convergence speed.

## Figures and Tables

**Figure 1 entropy-24-00641-f001:**
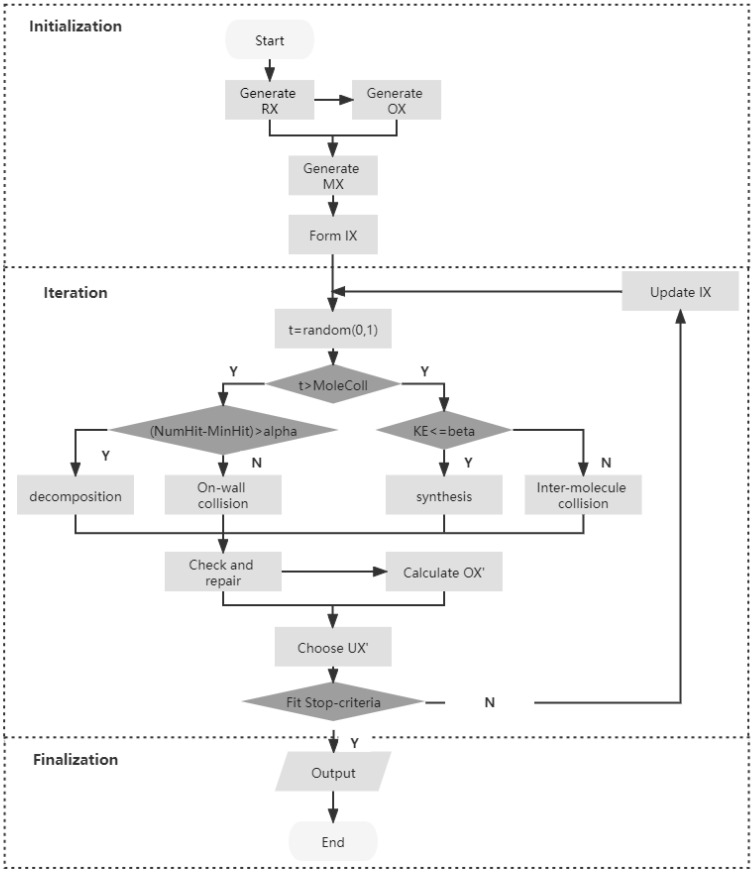
Framework of OBLIMCRO.

**Figure 2 entropy-24-00641-f002:**
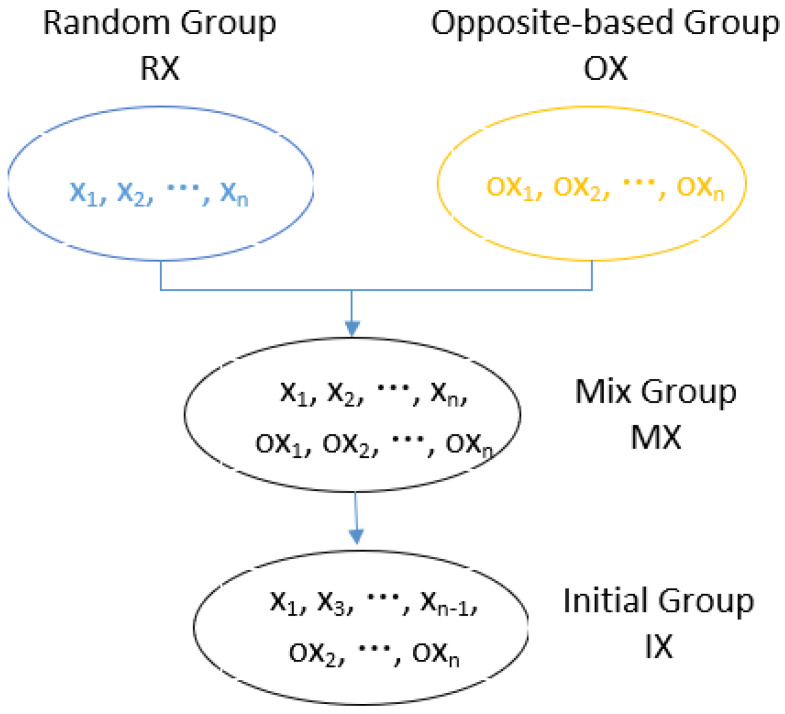
Initialization based on OBL.

**Figure 3 entropy-24-00641-f003:**
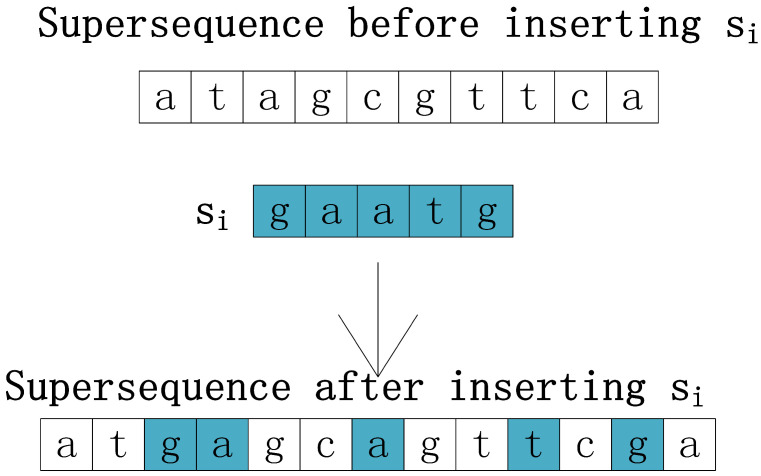
Population generation.

**Figure 4 entropy-24-00641-f004:**
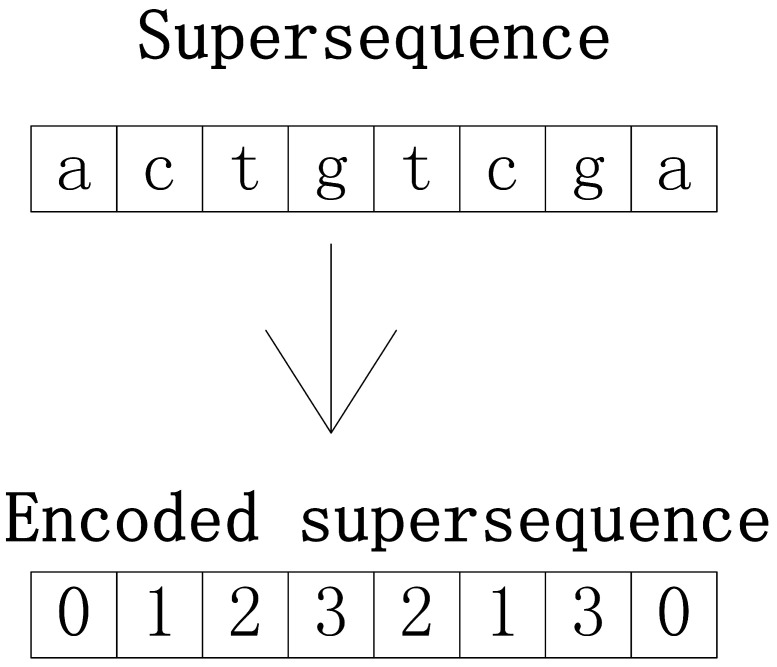
Solution representation.

**Table 1 entropy-24-00641-t001:** Average SCS Length in random datasets.

*n*	*k*	L					
		ACO	DR	IBS	CRO_SCS	IMCRO	OBLIMCRO
5	10	22.5	21.2	19.9	20.2	19.7	**19.6**
10	10	26.7	25.1	25.2	25.3	24.9	**24.8**
50	10	31.5	31.1	30.0	29.3	28.6	**28.6**
100	10	33.0	32.5	32.0	32.1	31.5	**30.5**
5	100	207.4	198.2	184.0	181.6	180.5	**179.8**
10	100	233.7	226.2	210.0	209.4	208.3	**207.6**
50	100	263.7	262.0	252.0	244.4	243.8	**241.4**
100	100	270.1	269.2	261.1	252.1	251.0	**247.3**
500	100	277.2	277.8	273.6	267.6	266.7	**266.5**
1000	100	281.7	278.5	276.8	270.1	269.7	**269.7**
5000	100	282.9	282.9	281.5	271.9	270.6	**269.9**
100	1000	2535.6	2531.6	2466.7	2443.2	2442.1	**2440.1**
500	1000	2565.6	2578.8	2540.2	2532.1	2530.5	**2530.2**
1000	1000	2570.8	2581.4	2555.5	2535.6	2533.9	**2531.4**
5000	1000	2590.6	2586.9	2571.6	2562.9	2561.8	**2558.7**

**Table 2 entropy-24-00641-t002:** Average execution time in random datasets.

*n*	*k*	T/s					
		ACO	DR	IBS	CRO_SCS	IMCRO	OBLIMCRO
5	10	0.8	0.018	0.03	0.008	0.008	**0.005**
10	10	1.00	0.033	0.03	0.03	0.03	**0.02**
50	10	2.3	0.1	0.07	0.08	0.065	**0.034**
100	10	3.5	0.15	0.12	0.08	0.055	**0.021**
5	100	5.9	0.6	0.14	0.02	0.013	**0.005**
10	100	8.6	1.18	0.22	0.14	0.12	**0.05**
50	100	16.3	4.07	0.46	0.37	0.26	**0.10**
100	100	23.5	7.28	0.91	0.65	0.52	**0.30**
500	100	65.5	27.3	3.06	1.69	0.92	**0.28**
1000	100	127.9	69.2	6.45	2.66	1.95	**0.56**
5000	100	706.6	339.4	41.65	5.01	5.01	**2.48**
100	1000	207.7	420.6	6.33	5.75	5.5	**2.1**
500	1000	651.1	1205.3	37.93	15.85	14.12	**6.43**
1000	1000	1296.5	2116.8	61.67	39.9	22.01	**9.56**
5000	1000	3101.6	3761.4	487.16	480.02	480.01	**238.31**

**Table 3 entropy-24-00641-t003:** Average SCS Length in real datasets.

Name	*n*	*k*	L					
			ACO	DR	IBS	CRO_SCS	IMCRO	OBLIMCRO
DNA-1	100	500	1346.9	1332.6	1280.7	1271.4	1271.0	**1270.0**
DNA-2	500	500	1520.0	1404.6	1352.7	1351.8	1350.8	**1349.7**
DNA-3	100	1000	2712.2	2670.1	2542.9	2442.5	2440.8	**2438.9**
DNA-4	500	1000	3092.1	2782.7	2664.4	2532.4	2530.3	**2530.2**
DNA-5	100	100	297.8	285.4	272.3	252.1	251.5	**250.6**
DNA-6	500	100	405.2	291.5	288.3	267.2	266.3	**266.2**
PROT-1	100	500	6908.2	4851.4	4349.7	4312.4	4311.6	**4310.5**
PROT-2	500	500	8910.4	5545.2	5229.3	5041.1	5040.8	**5028.2**
PROT-3	1000	500	11086	5748.7	5395.6	**5301.7**	5301.9	5302.1
PROT-4	100	100	1303.5	1005.6	**913.3**	920.9	921.2	921.8
PROT-5	500	100	1776.2	1205.1	**1107.6**	1126.0	1125.9	1126.7

**Table 4 entropy-24-00641-t004:** Average execution time in real datasets.

Name	*n*	*k*	T/s					
			ACO	DR	IBS	CRO_SCS	IMCRO	OBLIMCRO
DNA-1	100	500	151.3	349.4	3.00	1.91	1.91	**0.33**
DNA-2	500	500	613.1	540.6	17.14	8.35	7.83	**1.91**
DNA-3	100	1000	334.4	483.6	10.31	5.85	5.48	**1.43**
DNA-4	500	1000	1514.1	1156.5	42.4	15.7	15.46	**6.36**
DNA-5	100	100	41.97	7.87	0.92	0.56	0.49	**0.23**
DNA-6	500	100	92.0	37.55	2.95	0.81	0.73	**0.34**
PROT-1	100	500	560.3	1125.3	92.4	31.98	21.18	**8.90**
PROT-2	500	500	1450.2	1905.4	307.8	52.29	50.50	**24.01**
PROT-3	1000	500	3205.4	4002.7	1905.6	116.12	110.76	**62.46**
PROT-4	100	100	16.4	31.2	13.5	1.66	1.54	**0.94**
PROT-5	500	100	123.5	95.7	65.3	4.63	4.53	**2.69**

**Table 5 entropy-24-00641-t005:** Iterations in random datasets.

*n*	*k*	Threshold	CRO_SCS	IMCRO	OBLIMCRO
5	10	10	13	11	**10**
10	10	25	27	26	**25**
50	10	3	5	4	**4**
100	10	1	3	2	**2**
5	100	40	41	41	**40**
10	100	40	41	41	**41**
50	100	10	11	11	**11**
100	100	7	10	8	**8**
500	100	1	2	2	**2**
1000	100	1	2	2	**2**
5000	100	1	3	2	**1**
100	1000	55	57	56	**56**
500	1000	10	11	11	**11**
1000	1000	4	6	5	**5**
5000	1000	1	2	1	**1**

**Table 6 entropy-24-00641-t006:** Iterations in real datasets.

Name	*n*	*k*	Threshold	CRO_SCS	IMCRO	OBLIMCRO
DNA-1	100	500	20	205	201	**150**
DNA-2	500	500	6	61	61	**50**
DNA-3	100	1000	30	301	301	**240**
DNA-4	500	1000	16	161	161	**106**
DNA-5	100	100	3	6	4	**3**
DNA-6	500	100	41	45	41	**41**
PROT-1	100	500	40	402	401	**205**
PROT-2	500	500	8	82	81	**71**
PROT-3	1000	500	4	43	41	**33**
PROT-4	100	100	5	52	51	**35**
PROT-5	500	100	2	21	21	**10**

## Data Availability

Not applicable.

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
