# Peer review of "An Opposition-Based Learning CRO Algorithm for Solving the Shortest Common Supersequence Problem"

_entropy, 2022, doi:10.3390/e24050641_

Round 1

Reviewer 1 Report

A file has been attached.

Reviewer 2 Report

The article is well structured and presents an interesting novelty.

The presentation can be adjusted by:

  • At the beginning of the paper, the authors should write a deciphering of the abbreviation NP-hard. This abbreviation is not deciphered anywhere.
  • If there is a possibility in the paragraph "The 4 operators in this stage 185 are the same as those defined in [12]" in line 185 to briefly describe 4 operators, which were presented in the previous work, without resorting to exaggerating the allowable size of the article, it would be convenient for the reader.

The results of the research are presented by the authors very clearly. The results are particularly well described in comparison with the use of other algorithms, which allows readers to be convinced of the effectiveness of the proposed method

Reviewer 3 Report

The paper is well organised and proposes a new method to solve the shortest common supersequence problem, even if the application under consideration in the results section could have been clearer. This needs to be addressed.

The links 1 and 2 given into the data at the end of page 12 are not working: http://www.biomedcentral.com/content/supplementary/1471-2105-7-S4-S12-S1.zip

English spelling also needs checking in some places, for example, line 267: "the reduce of average running time" , where the word reduce should read "reduction".

Replace "statistical significant" by "statistically significant" in line 287

About the p-value being smaller than 0.05, p-value is not always is the only factor to judge statistical significance even if it can be an important one. Additional information (likelihood ratios?) that justifies statistical significance should be given, especially since one of them (0.04) was very close to the usual 0.05 threshold.

Round 2

Reviewer 1 Report

If possible, cite more CRO related papers in relevant sub-sections.